# Radiographic Markers of Hip Dysplasia and Femoroacetabular Impingement Are Associated with Deterioration in Acetabular and Femoral Cartilage Quality: Insights from T2 MRI Mapping

**DOI:** 10.3390/jimaging11100363

**Published:** 2025-10-14

**Authors:** Adam Peszek, Kyle S. J. Jamar, Catherine C. Alder, Trevor J. Wait, Caleb J. Wipf, Carson L. Keeter, Stephanie W. Mayer, Charles P. Ho, James W. Genuario

**Affiliations:** 1Department of Orthopedics, University of Colorado School of Medicine, Aurora, CO 80045, USA; kyle.jamar@cuanschutz.edu (K.S.J.J.); catherine.alder@cuanschutz.edu (C.C.A.); trevor.wait@cuanschutz.edu (T.J.W.); caleb.wipf@cuanschutz.edu (C.J.W.); stephanie.mayer@cuanschutz.edu (S.W.M.); charles.ho@cuanschutz.edu (C.P.H.); james.genuario@cuanschutz.edu (J.W.G.); 2UCHealth Steadman Hawkins Clinic, Englewood, CO 80112, USA; carson.keeter@cuanschutz.edu; 3Department of Radiology-Diagnostics, University of Colorado School of Medicine, Aurora, CO 80045, USA

**Keywords:** hip arthroscopy, hip dysplasia, femoroacetabular impingement, T2 mapping, MRI

## Abstract

Femoroacetabular impingement (FAI) and hip dysplasia have been shown to increase the risk of hip osteoarthritis in affected individuals. MRI with T2 mapping provides an objective measure of femoral and acetabular articular cartilage tissue quality. This study aims to evaluate the relationship between hip morphology measurements collected from three-dimensional (3D) reconstructed computed tomography (CT) scans and the T2 mapping values of hip articular cartilage assessed by three independent, blinded reviewers on the optimal sagittal cut. Hip morphology measures including lateral center edge angle (LCEA), acetabular version, Tönnis angle, acetabular coverage, alpha angle, femoral torsion, neck-shaft angle (FNSA), and combined version were recorded from preoperative CT scans. The relationship between T2 values and hip morphology was assessed using univariate linear mixed models with random effects for individual patients. Significant associations were observed between femoral and acetabular articular cartilage T2 values and all hip morphology measures except femoral torsion. Hip morphology measurements consistent with dysplastic anatomy including decreased LCEA, increased Tönnis angle, and decreased acetabular coverage were associated with increased cartilage damage (*p* < 0.001 for all). Articular cartilage T2 values were strongly associated with the radiographic markers of hip dysplasia, suggesting hip microinstability significantly contributes to cartilage damage. The relationships between hip morphology measurements and T2 values were similar for the femoral and acetabular sides, indicating that damage to both surfaces is comparable rather than preferentially affecting one side.

## 1. Introduction

Hip pathology, specifically femoroacetabular impingement (FAI) and hip dysplasia, has been shown to increase the rate of osteoarthritis in affected individuals [1,2]. These conditions affect a substantial (and growing) portion of the population, with asymptomatic radiographic hip dysplasia being found in up to 2.3% of the population and the incidence of FAI increasing by 80% in the last 25 years [3,4]. Young, active adults are predominantly affected, making early detection and intervention critical for long-term hip preservation [5].

Both FAI and developmental dysplasia of the hip are well-established precursors to early-onset osteoarthritis due to their abnormal biomechanical effects on the joint. In cam-type FAI, abnormal contact between the aspherical femoral head and acetabular rim leads to shear stress and progressive delamination of the acetabular cartilage, initiating early degenerative change [6]. Long-term population studies have confirmed that cam morphology significantly increases the risk of developing radiographic osteoarthritis compared to morphologically normal hips [7]. Similarly, dysplastic hip anatomy results in undercoverage of the femoral head and chronic microinstability, promoting uneven load distribution and cartilage wear. Untreated hip dysplasia accelerates osteoarthritic progression and increases the lifetime risk of total hip arthroplasty [8,9]. Collectively, these findings establish both FAI and dysplasia as key structural drivers of early hip joint degeneration.

Standing anteroposterior radiographs are often the initial imaging modality for assessing patients with hip pain where measures of joint space width may be used to detect osteoarthritis and assess severity. However, early signs of cartilage damage are not assessable on plain radiographs, presenting a challenge for early detection. Recent advancements in imaging techniques provide quantifiable data on cartilage health early in the process of pathologic change, with certain modalities able to detect differences in patients with early degenerative changes [10].

Magnetic resonance imaging (MRI) with T2 mapping evaluates cartilage health by detecting changes in the chondral collagen matrix and overall cartilage water content. When cartilage tissue degrades, the collagen architecture becomes less organized, leading to an increase in T2 relaxation time [11]. This technology reliably distinguishes between healthy and damaged cartilage tissues [12], where previous studies on acetabular and femoral articular cartilage specified ranges of 40–50 milliseconds (ms) for healthy cartilage and 50–60 ms for damaged cartilage [13,14,15]. T2 values are sensitive to subtle degeneration as demonstrated by Ho et al. who demonstrated location-dependent variation in T2 scores across individual subjects’ femoral heads and acetabula [14].

Radiographic measurements commonly used to evaluate hip morphology include lateral center-edge angle, Tönnis angle, acetabular version, alpha angle, and femoral neck-shaft angle [16,17]. Radiographic indicators of hip dysplasia, which predispose to microinstability, include a decreased lateral center-edge angle, an increased Tönnis angle, and decreased acetabular coverage, while signs of FAI include an increased alpha angle and acetabular retroversion [16,17]. The biomechanical differences between FAI and dysplasia are complex. Reductions in glucosaminoglucan content detected using dGEMRIC were detected in patients with hip dysplasia despite evidence of localized cartilage thinning [18]. FAI often causes focal injury at the impingement site along the acetabular rim [19]. Due to these differing mechanisms, it remains uncertain whether dysplasia or FAI causes more severe cartilage damage and whether the acetabular or femoral side is preferentially affected.

Previous studies investigating cartilage damage in dysplasia and FAI have largely relied on radiographs and conventional MRI sequences, both of which detect relatively advanced degeneration. These modalities often miss the earliest biochemical alterations in cartilage that precede gross defects. T2 mapping, by contrast, offers a non-invasive, quantitative technique that can reveal microscopic changes in collagen fiber orientation and hydration before they become apparent on traditional imaging modalities. By correlating radiographic morphology with these sensitive MRI biomarkers, clinicians may be able to stratify patients by risk of rapid cartilage deterioration and intervene before irreversible structural damage occurs. Such insights are especially relevant for young patients for whom delaying or preventing osteoarthritis has substantial implications for lifelong mobility and quality of life.

This study’s aims are as follows: (1) to evaluate whether dysplasia or FAI (measured radiographically) is more associated with cartilage damage (measured via T2 mapping), and (2) to evaluate if the femoral or acetabular cartilage is preferentially impacted in either condition.

## 2. Materials and Methods

### 2.1. Study Design and Sample

This is a single-center, multi-surgeon retrospective study of patients with symptomatic hip pathology treated between March 2021 and February 2023.

This cohort was reviewed as part of a previous study by the same authors [20]. The inclusion criteria consisted of patients who underwent primary hip arthroscopy surgery with labral repair or reconstruction, who were aged 14–50 years, who completed preoperative MRI with T2 mapping following the standard protocol specified below, and who completed a preoperative CT scan with 3D reconstruction, measured with Stryker HipMap software (HipMap, Stryker, Kalamazoo, MI, USA). Patients were excluded for a history of previous ipsilateral hip surgery, traumatic injury (acetabular fracture, hip dislocation, etc.), a history of inflammatory arthritis, a recent (within four weeks) history of an intra-articular injection of a steroid or bioactive agent, and patients who did not have appropriate T2 mapping imaging were excluded.

### 2.2. Preoperative Imaging Technique

Each patient included in this study underwent a CT scan with a Siemens Definition Flash Dual-Source scanner (Siemens Medical Solutions, Erlangen, Germany). Patients were imaged supine with helically acquired axial images obtained through bilateral hips at 2 mm intervals in bone algorithm, and reformatted in coronal, sagittal, and axial oblique planes (100 KVp, 81 reference mAs, 0.8 pitch, 0.5 s rotation, 128 × 0.6 mm collimation). Additionally, helical images are obtained through bilateral knees at 2 mm intervals in bone algorithm (100 KVP, 49 reference mAs, 0.8 pitch, 0.5 s rotation, 128 × 0.6 mm collimation). Three-dimensional surface-rendered images were also performed on an independent workstation for the pelvis (1 mm × 0.9 mm and 2 mm × 2 mm axial bone [Br59 kernel] and 2 mm × 2 mm axial soft tissue [Br40 kernel]; Coronal and Sagittal Multi-Planar Reformat views in 3 mm × 3 mm bone [Br59 kernel]) and bilateral knees (2 mm × 2 mm axial bone [Br59 kernel]). Field of view was modulated based on patient body habitus.

Each patient included in this study also completed MRI with a 3.0T MRI (Magnetom Vida, Siemens Medical Solutions, Erlangen, Germany) prior to hip arthroscopy. The imaging protocol included standard clinical morphologic sequences followed immediately by a sagittal multi-echo spine-echo (MESE) T2-mapping sequence. The T2 mapping sequence (TR/TE 1530.0/13.80–69.00 ms; VS, 0.5 × 0.5 × 3.0 mm^3^; Slices, 20; Slice thickness 3.0 mm; FOV, 160 mm; AT, 4:51 min; FOV read 150 mm; Flip Angle 180 degrees) was acquired in the sagittal plane for optimal mapping of the anterior lateral articular cartilage. The T2 mapping sequence was acquired after the standard MRI images were obtained and the patient had been recumbent for a prolonged period. The T2 mapping images were derived from the Siemens MapIT software algorithm (Seimens Healthineers, Erlangen, Germany).

### 2.3. Data Collection

The three independent, blinded reviewers analyzed T2 mapping on the optimal sagittal MRI cut using the Syngo.via T2 mapping software (Siemens Healthineers, Erlangen, Germany). Reviewers included an orthopedic surgery resident (A.P.) and two medical students (C.A., K.J.) supervised by a board-certified musculoskeletal radiologist (C.H.). For T2 analysis, the acetabular and femoral head cartilage were each divided into anterior, superior, and posterior zones. See Figure 1 and Figure 2 for a visual representation of this division in the mapping software. The average T2 mapping value for each respective zone on the acetabulum and femoral head were measured two times by each of the three reviewers and the average of these values was taken. Intraclass correlation values were obtained to confirm appropriate inter-observer agreement. Hip morphology measurements were recorded from CT scans using 3D reconstruction and measurement software (HipMap, Stryker, Kalamazoo, MI, USA). Measurements included alpha angle, lateral central edge angle (LCEA), acetabular coverage, femoral version, acetabular version at 12 o’clock, 2 o’clock, and 3 o’clock, femoral neck-shaft angle (FNSA), and Tönnis angle.

### 2.4. Statistical Analysis

Univariate Mixed Linear Models (MLMs) were used to assess the association between various radiographic measures and T2 mapping values. To account for the multiple measurements within each patient, the random effects of rater, zone number, measurement number, and pass number were considered. Regression coefficients (β) with corresponding 95% confidence intervals (CIs) and *p*-values were reported. To facilitate a direct comparison of effect sizes across different predictors, all continuous variables were separately centered and scaled and standardized regression coefficients (β*) were also reported.

Intraclass correlation (ICC) values were calculated for the mapping values. There was excellent agreement between the raters at the acetabular cartilage (ICC2k = 0.936, 95% CI [0.769, 0.974], *p* < 0.001) and femoral cartilage (ICC2k = 0.936, 95% CI [0.769, 0.974], *p* < 0.001).

Estimated marginal trends were examined to evaluate differences in the trends of LCEA, Tönnis grade, and alpha angle across anatomical zones and to determine whether these trends varied by zone. Additionally, estimated marginal means were calculated to compare T2 mapping values across zones while adjusting for the respective predictor variable within each model. Pairwise comparisons were adjusted using Tukey’s method. Statistical significance was set at α = 0.05. All statistics were performed in the statistical software, R (version 4.4.3, Vienna, Austria) [21].

## 3. Results

Forty-four hips from 36 patients qualified for enrollment into this study. Mean participant age was 31.5 years (range 14–46), mean body mass index was 24.6 kg/m^2^. A total of 58% of the analyzed hips were from female patients and 55% of the analyzed hips were right-sided. Eight patients (22%) received arthroscopic procedures on both hips.

The Univariate Mixed Linear Model results are summarized in Table 1 and Table 2. The standardized regression coefficients demonstrated that each radiographic parameter had a similar magnitude of impact (β value) on femoral and acetabular cartilage. An increased LCEA (β = −0.82, *p* < 0.001), acetabular coverage (β = −0.82, *p* < 0.001), FNSA (β = −0.70, *p* < 0.001), and alpha angle (β = −0.97, *p* < 0.001) were associated with improved cartilage quality (lower T2 values), where the β value represents the change in T2 mapping value per unit increase in the radiographic parameter of interest. Increased acetabular version at 12 o’clock (β = 0.67, *p* < 0.001), 2 o’clock (β = 0.64, *p* < 0.001), and 3 o’clock (β = 0.81, *p* < 0.001), and an increased Tönnis angle (β = 0.64, *p* < 0.001) were associated with decreased cartilage quality (Figure 3 and Figure 4).

Combined version demonstrated a positive association with acetabular cartilage T2 values (β = 0.39, *p* = 0.002) but demonstrated a smaller effect with femoral cartilage values (β = 0.24, *p* = 0.004). Femoral torsion did not have a statistically significant linear relationship with T2 mapping values for femoral or acetabular cartilage (*p* = 0.20 and *p* = 0.70, respectively).

## 4. Discussion

Significant associations were observed for acetabular and femoral articular cartilage and all morphology parameters except femoral torsion. These findings indicate that the patterns of articular cartilage wear are complex and dependent on multiple anatomic abnormalities. The data show that increased T2 mapping values, indicative of poorer tissue quality, were associated with a decreased LCEA, increased acetabular version at 12 o’clock, 2 o’clock, and 3 o’clock, an increased Tönnis angle, decreased acetabular coverage, and a decreased alpha angle. These values align with the described ranges of hip dysplasia [22], suggesting that hip dysplasia may predispose patients to more severe cartilage damage than other hip pathologies such as FAI.

The dysplastic bony anatomy of the hip joint leads to microinstability, which results in increased and non-uniform loading across the hip joint [19,23,24]. This altered loading pattern results in global cartilage quality deterioration through a biologically mediated inflammatory process rather than pure mechanical wear [18,25]. This phenomenon may help explain the discrepancy between the T2 MRI findings of diffuse femoral and acetabular cartilage damage versus the more localized mechanical wear noted on hip arthroscopy patients, which is typically more pronounced on the acetabulum. Since T2 mapping MRI measures changes at the molecular level, it can detect early, global signs of change prior to the development of localized lesions [26].

The results of this study confirm the impact of hip dysplasia on global cartilage quality, demonstrating that with an increased severity of hip dysplasia based on established radiographic markers, there is an associated universal decrease in acetabular and femoral cartilage quality based on T2 mapping MRI findings.

The associations between hip morphology measurements and T2 values were similar for the femoral and acetabular sides, suggesting the damage to both sides of the joint is comparable. The standardized regression coefficients (β) showed similar magnitudes for each radiographic parameter, whether assessing acetabular or femoral cartilage damage, suggesting that cartilage damage as a result of dysplasia or FAI does not preferentially favor one side of the joint over the other.

This study is not without limitations. Its sample size of 44 hips is relatively small, largely due to preoperative imaging requirements, though this number of patients was sufficient to detect statistically significant results for nearly every radiographic parameter measured. Larger future studies may allow for a stratification of the results and subgroup analyses to investigate if age, sex, or other patient factors have an impact on the relationships observed in these results. Regarding data collection, variability in image quality and some images showing extensive cartilage damage made T2 mapping identification challenging. Three independent reviewers and a regression modeling system accounting for variability through random effects help address this limitation, though the potential for inaccurate T2 mapping data remains. Another limitation of this study is its retrospective design, which inherently introduces the possibility of selection bias. The patients included were those who underwent both preoperative CT and MRI with T2 mapping as part of their clinical care, which may represent a subset of individuals with a more complex or symptomatic pathology. As a result, the findings may not be fully generalizable to the broader population of patients with radiographic dysplasia or FAI who are managed nonoperatively or who present earlier in the disease process.

Specific to the collection of T2 mapping data, another limitation is the potential effect of variability in patient positioning and the timing of MRI acquisition relative to daily activity. T2 relaxation times can be influenced by loading history, hydration status, and even diurnal variation, meaning that subtle differences in patient activity prior to imaging could contribute to variability in the measured values [11,12,13,14,15]. Although all scans were performed using standardized protocols after patients had been supine for a period of time, these physiological factors could not be fully controlled. Future investigations may benefit from a more rigorous standardization of pre-imaging activity or repeated imaging at different time points to assess the reproducibility of T2 values in the same patient.

An important future direction will be to investigate whether surgical correction of underlying structural abnormalities can meaningfully alter the trajectory of cartilage quality deterioration detected on T2 mapping. Although hip preservation procedures such as periacetabular osteotomy (PAO) for dysplasia and femoral or acetabular osteoplasty for FAI have been shown to relieve symptoms and improve functional outcomes [27,28,29], it remains unclear if these interventions halt or even reverse the biochemical changes within the cartilage matrix. Prospective trials that measure preoperative and serial postoperative T2 values could help clarify whether mechanical correction reduces abnormal loading sufficiently to stabilize cartilage microstructure, or whether biochemical degeneration progresses regardless of bony realignment. Such information would have major clinical implications, as demonstrating that cartilage quality can be preserved or restored with timely surgical intervention would provide a powerful justification for early treatment in young, symptomatic patients. It may also help us understand why some patients do better than others clinically after hip preservation surgeries. Furthermore, evaluating the differential response of femoral versus acetabular cartilage to surgical correction may offer insight into whether one surface is more amenable to recovery than the other. Ultimately, defining the capacity of cartilage showing microstructural degenerative changes to recover following surgical intervention will be essential for developing evidence-based treatment algorithms aimed not only at symptom relief but also at altering the natural history of hip joint degeneration.

## 5. Conclusions

Nearly all the radiographic parameters of patients’ bony anatomy on preoperative CT scans were associated with the T2 mapping values of femoral head and acetabular cartilage. Acetabular and femoral head cartilage were impacted equally by each radiographic parameter—there was no evidence that any particular bony anatomy preferentially impacts femoral versus acetabular cartilage. This study provides quantitative evidence that dysplastic morphology has a stronger association with global cartilage deterioration than FAI-related parameters, though these findings must be interpreted in the context of a single-institution, retrospective review. Future research should focus on longitudinal studies tracking T2 changes over time and investigating whether morphological corrections result in measurable cartilage preservation, which could inform evidence-based joint preservation strategies.

## Figures and Tables

**Figure 1 jimaging-11-00363-f001:**
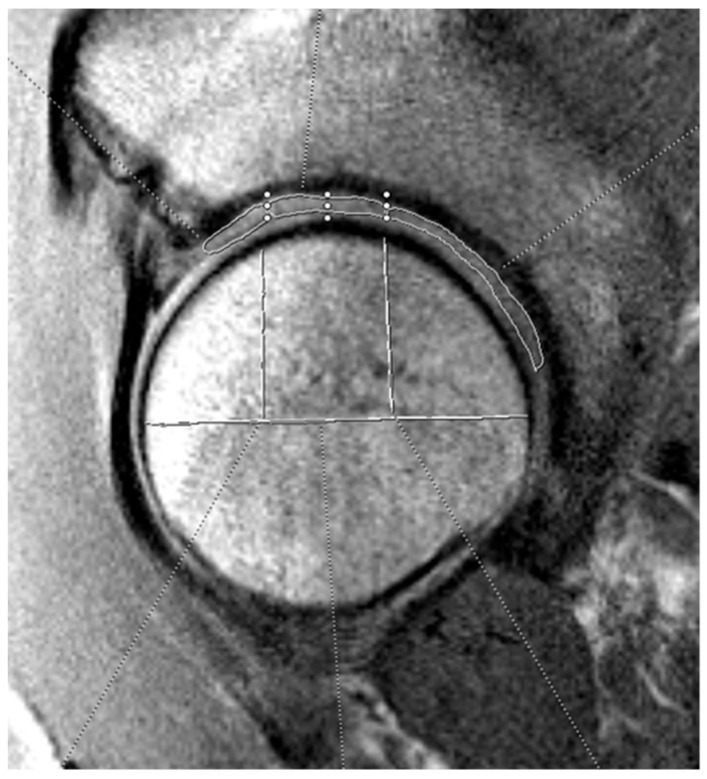
Example of acetabular cartilage T2 mapping on a sagittal MRI. The measurements were divided into three zones (anterior, superior, and posterior). The horizontal white line bisects the femoral head, and the vertical lines split the joint surface into 3 zones: anterior (left), superior (middle), and posterior (right). The three zones on the acetabular cartilage are drawn by free-hand technique based on these divisions.

**Figure 2 jimaging-11-00363-f002:**
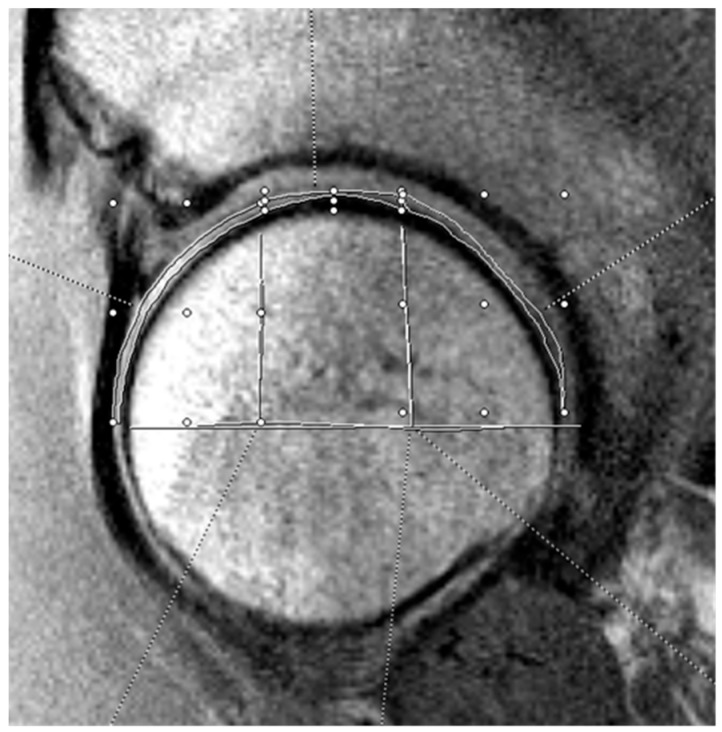
Example of femoral head cartilage T2 mapping on a sagittal MRI. The measurements were divided into three zones (anterior, superior, and posterior). The horizontal white line bisects the femoral head, and the vertical lines split the joint surface into 3 zones: anterior (left), superior (middle), and posterior (right). The three zones on the femoral head cartilage are drawn by free-hand technique based on these divisions.

**Figure 3 jimaging-11-00363-f003:**
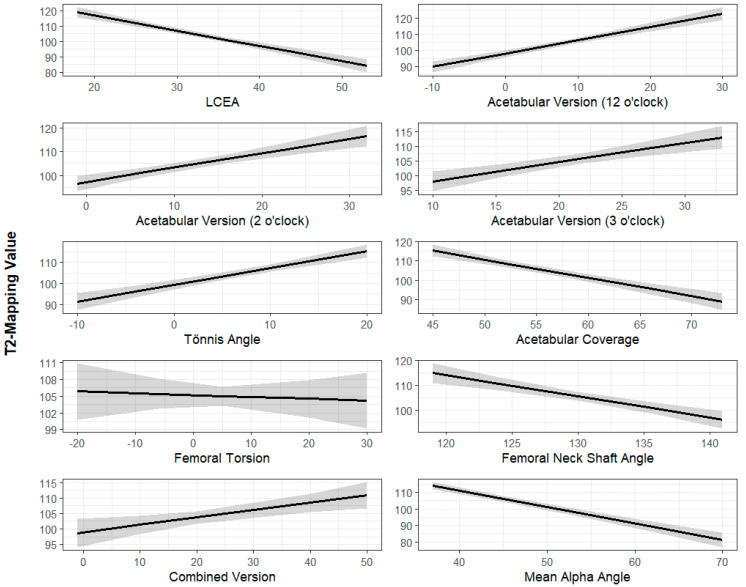
Radiographic parameters (x-axis) plotted against T2 mapping scores of femoral head cartilage (y-axis). The dark line represents the best-fit linear regression line while the shaded area represents the 95% confidence interval. All radiographic parameters are measured in degrees except for acetabular coverage, which is measured as a percentage. T2 mapping values are measured in milliseconds.

**Figure 4 jimaging-11-00363-f004:**
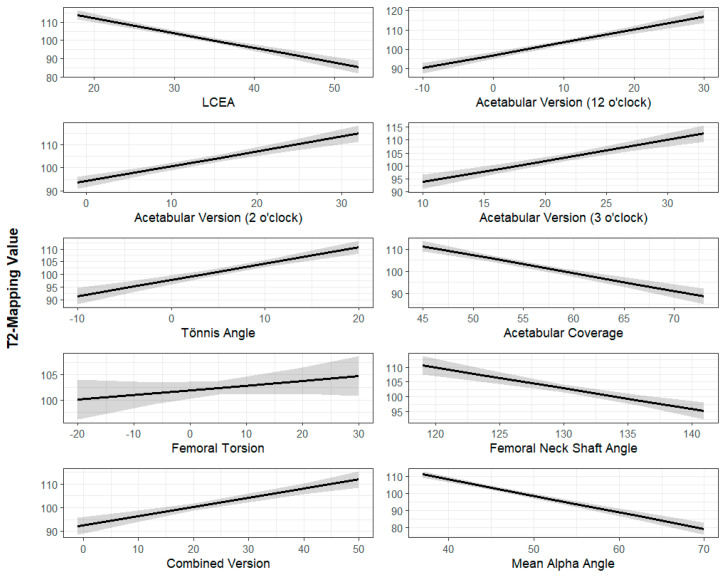
Radiographic parameters (x-axis) plotted against T2 mapping scores of acetabulum cartilage (y-axis). The dark line represents the best-fit linear regression line while the shaded area represents the 95% confidence interval. All radiographic parameters are measured in degrees except for acetabular coverage, which is measured as a percentage. T2 mapping values are measured in milliseconds.

**Table 1 jimaging-11-00363-t001:** Univariate associations of acetabular cartilage T2 values and hip radiographic measurements.

Model	Beta	95% CI	Std. Beta	*p*-Value
LCEA	−0.82	(−0.98, −0.66)	−0.20	<0.001
Acetabular version				
12:00	0.67	(0.53, 0.81)	0.17	<0.001
2:00	0.64	(0.48, 0.81)	0.12	<0.001
3:00	0.81	(0.58, 1.04)	0.13	<0.001
Tönnis Angle	0.64	(0.47, 0.81)	0.16	<0.001
Acetabular coverage	−0.82	(−0.01, −0.63)	−0.18	<0.001
Femoral torsion	0.09	(−0.06, 0.24)	0.03	0.20
FNSA	−0.70	(−0.94, −0.45)	−0.10	<0.001
Combined version	0.39	(0.26, 0.52)	0.12	0.002
Alpha Angle	−0.97	(−1.11, −0.83)	−0.24	<0.001

**Table 2 jimaging-11-00363-t002:** Univariate associations of femoral head cartilage T2 values and hip radiographic measurements.

Model	Beta	95% CI	Std. Beta	*p*-Value
LCEA	−0.99	(−1.18, −0.80)	−0.19	<0.001
Acetabular version				
12:00	0.83	(0.66, 1.00)	0.17	<0.001
2:00	0.61	(0.40, 0.81)	0.09	<0.001
3:00	0.66	(0.38, 0.94)	0.08	<0.001
Tönnis Angle	0.79	(0.58, 1.00)	0.16	<0.001
Acetabular coverage	−0.95	(−1.19, −0.71)	−0.17	<0.001
Femoral torsion	−0.03	(−0.22, 0.16)	−0.01	0.70
FNSA	−0.85	(−1.15, −0.54)	−0.10	<0.001
Combined version	0.24	(0.08, 0.40)	0.06	0.004
Alpha Angle	−1.00	(−1.17, −0.83)	−0.20	<0.001

## Data Availability

The original contributions presented in this study are included in the article. Further inquiries can be directed to the corresponding author.

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
