# Peer review of "Radiographic Markers of Hip Dysplasia and Femoroacetabular Impingement Are Associated with Deterioration in Acetabular and Femoral Cartilage Quality: Insights from T2 MRI Mapping"

_2313-433X, 2025, doi:10.3390/jimaging11100363_

Round 1

Reviewer 1 Report

Comments and Suggestions for Authors

This study aims to evaluate the relationship between hip morphology measurements from CT scans and T2 mapping values of hip articular cartilage assessed by MRI. The authors conclude that abnormal hip morphology, indicative of dysplasia (decreased LCEA, increased Tönnis angle, decreased acetabular coverage), was associated with increased cartilage damage (increased T2 value).

The structures of methodology are robust and well-explained. The statistic analysis is appropriate. The discussion is concise and relative to the derived results. The inherited bias from retrospectively-based study such as selection bias, sensitivity of MRI T2 value acquisition to physiologic status had been well explained in limitation. Overall, the manuscript is nearly ready for publish.

 However, there are still some issues can be addressed to add strength.

  1. Lack of a demographic data which provides clinical diagnosis (FAI or hip dysplasia), age, gender and BMI. Since the authors suggest (Discussion, Line 204-205) that hip dysplasia may predispose patients to more severe cartilage damage than FAI, numbers of patients diagnosed clinically as FAI included in this study should be provided. Besides, age, gender and BMI are all important factors contributing to arthritis.
  2. All the patients were symptomatic hip indicated for arthroscopic labral repair. However, the status of labral lesions was not described and analyzed as an important variable in this study.
  3. Subregional anatomical evaluation of T2 values for femoral head and acetabulum is technically feasible. It would be very helpful to provide clues to answer the first study aim.

In summary, the manuscript presents valuable insights into the relationship between hip morphology and cartilage quality in patients with FAI or dysplasia. Addressing the potential limitations and expanding on the clinical significance of the findings would further strengthen the manuscript and enhance its impact.

Author Response

Thank you for your thorough and thoughtful review of our manuscript. See below for a point by point response to your suggestions:

  1. Lack of a demographic data which provides clinical diagnosis (FAI or hip dysplasia), age, gender and BMI. Since the authors suggest (Discussion, Line 204-205) that hip dysplasia may predispose patients to more severe cartilage damage than FAI, numbers of patients diagnosed clinically as FAI included in this study should be provided. Besides, age, gender and BMI are all important factors contributing to arthritis.

The authors agree that the demographic data is vital to this study. Age, sex, and BMI statistics have been added to the results section. Regarding the clinical diagnosis of FAI versus dysplasia, these numbers were not included in the manuscript because of a large number of patients with mixed pathology (for example, patient’s with decreased LCEA, but also an increased alpha angle, suggestive of both cam-type FAI and dysplasia). Given this, the intent of the analysis was to focus on individual radiographic parameters rather than grouping patients into specific diagnoses of FAI or dysplasia.

2. All the patients were symptomatic hip indicated for arthroscopic labral repair. However, the status of labral lesions was not described and analyzed as an important variable in this study.

The authors wholeheartedly agree that the labrum is a key piece of this study. As such, the labrum was analyzed separately, including T2 mapping MRI measurements, size (as measured on MRI) and compared to similar radiographic parameters as done in this study, but published as a separate manuscript (citation below). It was the authors’ intention to focus solely on the impact of bony anatomy on the articular cartilage in this study.

Peszek A, Alder CC, Jamar K, Wait TJ, Wipf CJ, Keeter CL, Mayer SW, Ho CP, Genuario JW. Labral size measured on preoperative magnetic resonance imaging not predictive of the need for labral reconstruction in patients undergoing primary hip arthroscopy. J Hip Preserv Surg. 2024 Dec 10;12(1):20-26. doi: 10.1093/jhps/hnae043. PMID: 40331069; PMCID: PMC12051866.

3. Subregional anatomical evaluation of T2 values for femoral head and acetabulum is technically feasible. It would be very helpful to provide clues to answer the first study aim.

The authors agree that this analysis would be quite useful if able to be done properly. In our study, a significant number of patients had mixed pathology (i.e. dysplastic features with cam-type and/or pincer-type femoroacetabular impingement). Cartilage wear patterns would be expected to be different in these different pathology types, thus the most fruitful way to carry this out would be to compare patients with isolated pathology (i.e. only dysplasia or only one type of FAI) and compare the patterns between groups. Doing so would leave a very small number of patients in each group, underpowered to detect a meaningful difference. Carrying out the analysis without separating patients based on pathology would give us a heterogeneous data pool which was be difficult to draw meaningful clinical conclusions from. For these reasons, this analysis was not carried out and included in the manuscript, and the data from the three zones on the femur and acetabulum were instead pooled to give a more global evaluation of each cartilage surface.

Reviewer 2 Report

Comments and Suggestions for Authors

This manuscript studied relationship between hip morphology measurements collected from three-dimensional (3D) reconstructed computed tomography (CT) scans and T2 mapping values of hip articular cartilage. This topic should be of interest, but some revisions are needed to improve and clarify the merit of this manuscript as follows:

-Introduction: 3D CT morphology measurements and T2 mapping values are generally and strongly related, as the underlying mechanical causes detected by CT often explain the biochemical damage in the cartilage measured by T2 mapping. CT provides crucial information about bony abnormalities that create areas of abnormal stress, while T2 mapping assesses the resulting cartilage health. The two techniques are complementary and are often used together for a comprehensive assessment of the hip joint. The review on the use of such techniques in femoroacetabular impingement (FAI) and hip dysplasia (DDH), where bony abnormalities are known to cause or contribute to cartilage damage.  Authors may further review the related previous studies in the context of such techniques and their correlations.

-Authors should acknowledge and made citation that this study was already published as a conference paper in J Hip Preserv Surg. 2025 Mar 27;12(Suppl 1):i84. doi: 10.1093/jhps/hnaf011.271.

Would this manuscript and that one published on be similar or different in what aspect?

-Due to the nature of a retrospective  study, the description of preoperative imaging technique should be written as such. At present, it is rather like a prospective study methodology.

-It is not clear whether calibration among the three independent evaluators was conducted to ensure consistency and inter-rater reliability. Calibration is important to minimize subjective variation and improve accuracy in assessments. Please specify whether this process was performed and describe how inter-person agreement was evaluated. In addition, please  indicate which of the authors served as the evaluators.

-Statistical analysis: Please indicate the software, version, manufacture used for analysis.

-The statement that the study  "This study is not without limitations. Its sample size of 44 hips is relatively small, largely due to pre-operative imaging requirements, though it was sufficiently powered to detect statistically significant results for nearly every radiographic parameter measured." may be misleading. Statistical significance does not inherently indicate adequate statistical power. Please clarify how the power analysis was performed and whether the sample size of 44 hips was determined a priori to be sufficient for detecting meaningful differences in all measured parameters.

-Conclusion: Since there are several limitations that affect the generalizability of this study, the conclusion should be revised. It should avoid definitive statements about advantages or findings and instead present the results more cautiously, emphasizing that they are preliminary and should be interpreted within the context of the study’s constraints. In addition, the claim of novelty may not be fully accurate, as such correlations have been described previously. The authors are encouraged to reconsider or reframe the novelty statement, ensuring it is supported by an appropriate comparison with existing literature.

-COA number should also be stated, not just a protocol number.

Author Response

Thank you for your thorough and thoughtful review of our manuscript. Your suggestions have been incorporated to strengthen out submission. See below for a point-by-point response to each suggestion:

"-Introduction: 3D CT morphology measurements and T2 mapping values are generally and strongly related, as the underlying mechanical causes detected by CT often explain the biochemical damage in the cartilage measured by T2 mapping. CT provides crucial information about bony abnormalities that create areas of abnormal stress, while T2 mapping assesses the resulting cartilage health. The two techniques are complementary and are often used together for a comprehensive assessment of the hip joint. The review on the use of such techniques in femoroacetabular impingement (FAI) and hip dysplasia (DDH), where bony abnormalities are known to cause or contribute to cartilage damage.  Authors may further review the related previous studies in the context of such techniques and their correlations."

Previous studies that have looked at hip morphological changes may impact T2 mapping have been referenced and discussed. Additional details of the macroscopic associations between CT/radiographic bony morphology and articular cartilage wear have been added added as well to make for a more complete background discussion.

"-Authors should acknowledge and made citation that this study was already published as a conference paper in J Hip Preserv Surg. 2025 Mar 27;12(Suppl 1):i84. doi: 10.1093/jhps/hnaf011.271.

Would this manuscript and that one published on be similar or different in what aspect?"

The citation you reference is to an abstract of this work submitted to the International Society for Hip Arthroscopy (ISHA) 2024 conference. This work was presented as a poster at that conference, but the submitted manuscript has never been published.

"-Due to the nature of a retrospective study, the description of preoperative imaging technique should be written as such. At present, it is rather like a prospective study methodology."

The wording of that section was thoroughly reviewed and updated to ensure all verbiage was in the past tense to reflect the retrospective nature of this. The protocols described are standard for the institution at which these imaging studies were performed, so the details are spelled out to add to the reproducibility of our findings.

"-It is not clear whether calibration among the three independent evaluators was conducted to ensure consistency and inter-rater reliability. Calibration is important to minimize subjective variation and improve accuracy in assessments. Please specify whether this process was performed and describe how inter-person agreement was evaluated. In addition, please indicate which of the authors served as the evaluators."

See Section 2.3 of the manuscript. The wording has been updated to address all of the above. See also section 2.4 for further details of intraclass correlation data.

"-Statistical analysis: Please indicate the software, version, manufacture used for analysis"

Data analysis was completed in the software, R (version 4.4.3). This has been added to the manuscript

"-The statement that the study  "This study is not without limitations. Its sample size of 44 hips is relatively small, largely due to pre-operative imaging requirements, though it was sufficiently powered to detect statistically significant results for nearly every radiographic parameter measured." may be misleading. Statistical significance does not inherently indicate adequate statistical power. Please clarify how the power analysis was performed and whether the sample size of 44 hips was determined a priori to be sufficient for detecting meaningful differences in all measured parameters."

We appreciate the reviewer’s comment and agree that statistical significance does not, by itself, confirm adequate statistical power. In this study, no formal a priori power analysis was performed; rather, the available sample (44 hips) was determined by the number of patients meeting inclusion criteria. Post hoc, we observed statistically significant effects, suggesting that the sample size was sufficient to detect meaningful effects for these outcomes, though we acknowledge that some analyses may have been underpowered to detect smaller effect sizes. The term “powered” was removed from the quoted sentence to avoid any confusion.

"-Conclusion: Since there are several limitations that affect the generalizability of this study, the conclusion should be revised. It should avoid definitive statements about advantages or findings and instead present the results more cautiously, emphasizing that they are preliminary and should be interpreted within the context of the study’s constraints. In addition, the claim of novelty may not be fully accurate, as such correlations have been described previously. The authors are encouraged to reconsider or reframe the novelty statement, ensuring it is supported by an appropriate comparison with existing literature."

The use of the word “novel” here was intended to refer to the quantitative data collected in this study, not necessarily the idea that dysplastic morphology is associated with cartilage wear. Nonetheless, we have removed the word for clarity’s sake. Your point that the conclusions are written to be a bit too definitive is well taken. Wording has been added to this section to clarify that these results should be interpreted in the context of this study’s limitations.

 "-COA number should also be stated, not just a protocol number."

The authors are not familiar with "COA" in this context and was not a part of our institution’s IRB process. Please let us know if there is additional information we can provide.

Reviewer 3 Report

Comments and Suggestions for Authors

The Article Radiographic Markers of Hip Dysplasia and Femoroacetabular Impingement are Associated with Deterioration in Acetabular and Femoral Cartilage Quality: Insights from T2 MRI Mapping.

Hip pathology, specifically femoroacetabular impingement (FAI) and hip dysplasia, has been shown to increase the rate of osteoarthritis in affected individuals.

Femoroacetabular impingement (FAI) and hip dysplasia have been shown to increase the risk of hip osteoarthritis in affected individuals.

The study aims to evaluate the relationship between hip morphology measurements collected from three-dimensional (3D) reconstructed computed tomography (CT) scans and T2 mapping values of hip articular cartilage assessed by three independent, blinded reviewers on the optimal sagittal cut.

The main question addressed by the research is clear.

The topic is original  and it add new information.

Materials and Methods:
This is a single-center, multi-surgeon retrospective study of patients with symptomatic hip pathology treated between March 2021 and February 2023. 

Preoperative Imaging Technique
Each patient underwent CT scan with a Siemens Definition Flash Dual-Source scanner 

Statistical Analysis: Univariate Mixed Linear Models (MLMs) were used to assess the association between various radiographic measures and T2 mapping values.

Conclusions:  This study provides novel quantitative evidence that dysplastic morphology has a stronger association with global cartilage deterioration than FAI related parameters.

The conclusions consistent with the evidence and arguments and  they address the main question. 

The references are appropriate.

The tables and figures clear ilustrated the topic.

Author Response

Thank you for your thoughtful review of our manuscript. Please note we have made some small changes to address the other reviewers’ comments, but no major changes to the substance of the paper.

Round 2

Reviewer 2 Report

Comments and Suggestions for Authors

The authors have revised the manuscript accordingly. However, one comment remains to be resolved. In the case of the certificate of approval (COA) number, IRB would generally issue a COA after protocol approval, with the COA number and approval date for reference. The COA number (doc number) is not the same as the protocol number. In general, we will refer to the COA number, not the protocol number. Please provide if there is. Not sure if the protocol code is the same as COA number in this case.

Author Response

Thank you for your addition review of our manuscript. After reviewing our own IRB further, it appears you are correct that protocol number and COA number are the same. Wording at the end of the manuscript has been changed to reflect this preferred nomenclature.